# The Role of Non-Enzymatic Degradation of Meropenem—Insights from the Bottle to the Body

**DOI:** 10.3390/antibiotics10060715

**Published:** 2021-06-14

**Authors:** Uwe Liebchen, Sophie Rakete, Michael Vogeser, Florian M. Arend, Christina Kinast, Christina Scharf, Michael Zoller, Ulf Schönermarck, Michael Paal

**Affiliations:** 1Department of Anesthesiology, University Hospital, LMU Munich, 81377 Munich, Germany; Uwe.Liebchen@med.uni-muenchen.de (U.L.); Christina.Kinast@med.uni-muenchen.de (C.K.); Christina.Scharf@med.uni-muenchen.de (C.S.); Michael.Zoller@med.uni-muenchen.de (M.Z.); 2Institute of Laboratory Medicine, University Hospital, LMU Munich, 81377 Munich, Germany; Sophie.Rakete@med.uni-muenchen.de (S.R.); Michael.Vogeser@med.uni-muenchen.de (M.V.); Florian.Arend@med.uni-muenchen.de (F.M.A.); 3Department of Medicine IV, University Hospital, LMU Munich, 81377 Munich, Germany; Ulf.Schoenermarck@med.uni-muenchen.de

**Keywords:** meropenem, open-ring metabolite (ORM), continuous infusion, stability, pharmacokinetic, isotope dilution liquid chromatography tandem mass spectrometry (ID-LC-MS/MS)

## Abstract

Several studies have addressed the poor stability of meropenem in aqueous solutions, though not considering the main degradation product, the open-ring metabolite (ORM) form. In the present work, we elucidate the metabolic fate of meropenem and ORM from continuous infusion to the human bloodstream. We performed in vitro infusate stability tests at ambient temperature with 2% meropenem reconstituted in 0.9% normal saline, and body temperature warmed buffered human serum with 2, 10, and 50 mg/L meropenem, covering the therapeutic range. We also examined meropenem and ORM levels over several days in six critically ill patients receiving continuous infusions. Meropenem exhibited a constant degradation rate of 0.006/h and 0.025/h in normal saline at 22 °C and serum at 37 °C, respectively. Given that 2% meropenem remains stable for 17.5 h in normal saline (≥90% of the initial concentration), we recommend replacement of the infusate every 12 h. Our patients showed inter-individually highly variable, but intra-individually constant molar ORM/(meropenem + ORM) ratios of 0.21–0.52. Applying a population pharmacokinetic approach using the degradation rate in serum, spontaneous degradation accounted for only 6% of the total clearance.

## 1. Introduction

Meropenem is a broad spectrum antibiotic with excellent activity against many pathogens that is used to treat a variety of bacterial infections [1]. As a beta-lactam antibiotic, meropenem exhibits a time-dependent antibacterial effect, i.e., the free concentration should be maintained above the minimum inhibitory concentration (MIC) of a target pathogen for the entire dosing interval (expressed as *f*T_>MIC_ = 100%) [2]. For complicated infections, some guidelines even recommend target trough levels with *f*T_>4-8x MIC_ = 100% to optimize clinical efficacy [3]. Recently, increasing concerns have been raised that meropenem exposure might be inadequate, especially in critically ill patients, with the necessity for therapeutic drug monitoring (TDM)-guided individualized treatment [4,5,6].

In order to maximize antimicrobial target attainment and clinical cure in severe illness, meropenem is typically administered with continuous infusion regimens after application of a loading dose [7,8,9]. The administration of meropenem in outpatient parenteral antimicrobial therapy (OPAT), representing the administration of IV antibiotics without hospitalization, is also becoming increasingly popular in clinical practice [10]. However, continuous infusion of meropenem has been limited owing to concerns with instability, including prolonged storage time before administration.

According to the European and United States Pharmacopoeia, stability of an infusate is only given when the drug concentration remains above 90% of the initial concentration during the entire infusion interval [11,12]. Numerous studies have investigated the stability of meropenem in aqueous solutions at various conditions, including antibiotic concentration, temperatures and pH values [13,14,15,16,17,18,19,20,21,22,23]. All these studies focused solely on in vitro stability data. However, they do not examine meropenem degradation in vivo and they also do not investigate the formation of the open-ring metabolite (ORM) of meropenem, the main degradation product formed by hydrolysis of the beta-lactam ring [24]. In vivo, approximately 70% of meropenem is excreted by the kidneys, while approximately 30% is non-renally excreted. The high proportion of renal excretion necessitates dose adjustments based on the kidney function [25]. In addition, reporting of the ORM concentration is of added value for TDM purposes, given that unusually high ORM levels may be indicative of pre-analytical issues (e.g., improper sample handling with in vitro meropenem degradation).

The aim of the present study was thus the comprehensive assessment of meropenem stability, continuously tracking both meropenem and the ORM from the infusate to the in vivo condition by isotope dilution liquid chromatography tandem mass spectrometry (ID-LC-MS/MS). For this purpose, extensive in vitro stability testing was performed and supplemented by the investigation of serum samples from patients who received meropenem as a continuous infusion. A population pharmacokinetic approach was employed to investigate the proportion of spontaneous decay to the total elimination.

## 2. Results

### 2.1. Infusate Stability at Room Temperature

Meropenem dissolved with 2% in normal saline solution decreased continuously with a degradation rate of 0.006/h (=percentage decay per hour) at 22 °C, with a recovery of 86.6% (% of the initial concentrations, t = 0 h) after 24 h. After 17.5 h, the stability limit of 90% as proposed by the European and United States Pharmacopoeia was crossed. The ORM increased at the same time, but was not quantitatively recovered as a metabolite (see Figure 1). The concentration of meropenem in mg/L and the ORM in normal saline at each time point is visualized in Figure 1.

### 2.2. Stability in Serum

No concentration-dependent degradation effect was observed in buffered serum heated to 37 °C. Instead, uniform degradation rates of 0.025/h were obtained for 2, 10, and 50 mg/L total meropenem with a recovery of 54.9% after 24 h. As observed in normal saline, the ORM increased at the same time, but was not quantitatively recovered as a metabolite (see Figure 2). The degradation of meropenem and the increase of ORM in 37 °C warmed buffered serum are illustrated in Figure 2.

### 2.3. Pharmacokinetics in Patients

A total of 24 serum samples from 6 patients (4 male, 2 female) were included in the study (see Table 1). The median age was 46 years (range: 35–72 years), the median body weight was 86 kg (range: 47–170 kg), and the median meropenem infusion rate was 6 g/24 h (range: 3–6 g/24h). The glomerular filtration rates were 28–307 mL/min (median 139 mL/min). None of the patients received extracorporeal renal replacement therapy.

Over several days of continuous meropenem administration, all patients showed stable intra-individual meropenem and ORM steady state concentrations, but highly variable inter-individual ORM/(meropenem + ORM) metabolic ratios of 0.21–0.52 (median 0.28); see Figure 3. The ORM/(meropenem + ORM) ratio was correlated with the GFR (R^2^ 0.46, *p* < 0.001). Patient no. 2, with the highest renal impairment (median GFR 42 mL/min), also had the highest metabolic ratio. The median meropenem concentration was 19.74 mg/L (range: 7.25–31.25 mg/L) and the median ORM concentration was 7.73 mg/L (range: 2.71–23.37 mg/L).

The one-compartment model with first-order elimination, inter-individual variability on the total clearance, and a proportional residual variability adequately described the meropenem concentrations (see Table 2 and Figure 4 and Figure 5). The total clearance was 11.4 L/h (range: 5.2–25.3 L/h). The glomerular filtration rate clearance (CL_GFR_) represented the largest clearance fraction with 7.1 L/h (62% of total clearance). The unexplained residual clearance (CL_nonGFR_) was 3.6 L/h (32%) and spontaneous decay (CL_decay_) accounted for about 0.7 L/h clearance (6% of total clearance).

## 3. Discussion

In the present study, we investigated the degradation of meropenem both in vitro and in vivo, applying a high precision isotope dilution LC-MS/MS method designed to quantify both meropenem and its main degradation product, the open-ring metabolite (ORM) form, which is microbiologically inactive. The inaccuracy and imprecision of the quantitative assay were consistently ≤8% (in most cases < 5%), which allows reliable conclusions to be drawn.

Several studies have shown that meropenem is relatively unstable after reconstitution in aqueous solution. Accordingly, delivery by continuous infusion over 24 h is generally considered unacceptable. Consistent with previous findings by Manning et al. [27], our results show that normal saline solutions with the standard concentration of 2% meropenem in normal saline retain >90% of their initial concentration for 17.5 h and 86.6% for 24 h at 22 °C, respectively. The ORM concentration is not increasing equimolarly with the degradation of meropenem, indicative of further conversion of the ORM (e.g., by decarboxylation) [24]. For clinical practice and in agreement with previous stability studies, we recommend the administration of meropenem prepared in two separate continuous 12 h infusions, respectively [27,28,29,30].

Meropenem stability has also been shown to be influenced by the drug concentration. Degradation rates of reconstituted meropenem increase with higher concentrations in infusates (mg/mL range) [22,27,31], which can be attributed to intermolecular aminolysis by a nucleophilic attack of one meropenem molecule opening the beta-lactam ring of a second molecule of meropenem. Concentration-dependent degradation in the mg/L scale is also evident in frozen human plasma samples [32]. To the best of our knowledge, there are no studies that simultaneously investigate the degradation of meropenem and its conversion to ORM in vivo using a single analytic method. In the present study, we used meropenem-fortified human serum at body temperature as an approximate in vitro model for the human bloodstream. To avoid sample alkalization and enhanced meropenem degradation due to loss of dissolved carbon dioxide, the serum pH was stabilized with phosphate buffer. The three tested concentrations of 2, 10, and 50 mg/L that are representative of the therapeutic range produced identical meropenem degradation rates of 0.025/h during the 24 h interval tested with a non-equimolar increase of the ORM. This degradation rate yielded a half-life of about 27.7 h, which is significantly longer than previously described by Harrison et al. with approximately 11 h [33]. The longer in vitro half-life of meropenem in our experiments might be explained by the fact that we stabilized the pH value by buffer addition. Our test conditions are closest to physiological in vivo conditions and should thus be considered the most valid.

We also investigated the relationship of meropenem and the ORM in vivo in patients receiving continuous infusion of meropenem. The formation of the ORM reflects the non-renal elimination in vivo and can either be caused by spontaneous, non-enzymatic beta-lactam opening or enzymatic hydrolysis by renal dehydropeptidase-1 (DHP-1), although meropenem appears to be very stable against human DHP-1 [33,34]. In healthy individuals, meropenem is rapidly excreted unchanged in the urine (approximately 70% of the dose, t_1/2_ ≈ 1 h) and the remainder mainly by conversion to the ORM (approximately 30% of the dose, t_1/2_ ≈ 2–3 h) [35]. In subjects with impaired renal function, non-renal meropenem clearance via the ORM formation increases up to 50% [36,37]. In line with these findings, continuous infusion samples from our patients exhibited highly variable inter-individual metabolic ratios of ORM/(ORM + meropenem), with the highest ratio for patient number 2 with the most compromised renal function (median GFR: 42 mL/min). With timewise stable renal function, all patients presented with almost constant metabolic ratios over several days. Consequently, a single determination of ORM in routine clinical practice could be informative about the proportion of non-renal elimination in an individual patient and helpful for personalized dose adjustments. Total clearance in our population was slightly increased compared with previously published meropenem models in critically ill patients (7.7–9.4 L/h), which can be attributed to an overall hyperdynamic renal function (median GFR 139 mL/min) [26,38,39]. Our population pharmacokinetic model indicated a low impact of spontaneous degradation on the total clearance (6%), while glomerular filtration rate accounted for about 62%. Consequently, the remaining 32% of the clearance can be attributed to elimination by the renal DHP-1 and net tubular secretion, as mentioned earlier [33]. However, the mean percentage of 28% ORM indicates metabolization by renal DHP-1 rather than tubular net secretion, as the latter would not yield further ORM.

Our study has several limitations. The analysis of spontaneous degradation was an in vitro analysis and the implementation in a population pharmacokinetic model only represents an approximation of the in vivo condition. Our study would also have benefited considerably from the analysis of urine data. Unfortunately, these samples were not available for analysis. Still, our approach provides interesting and novel clues about the in vivo metabolism of meropenem. Furthermore, the determined elimination rates can be implemented in future model-based dose optimizations. In particular, physiologically-based pharmacokinetic models could benefit from integrating the determined elimination rates. Our population pharmacokinetic model was based on a reduced number of patients with sparse sampling and included fixed parameters. We would thus like to explicitly point out that this model should not be used for dose optimization strategies without restrictions.

In summary, the present study provides a deeper understanding about the stability of meropenem both in vitro and in vivo. Spontaneous degradation in serum accounts for only a small fraction of the non-renal elimination. Meropenem reconstituted at 2% in normal saline is reasonably stable at room temperature and, accordingly, requires only two separate 12 h infusions as a part of a 24 h continuous infusion regimen. Concomitant quantification of the open-ring metabolite (ORM) form could be helpful in dose adjustments in individual patients receiving meropenem therapeutic drug monitoring (TDM).

## 4. Materials and Methods

### 4.1. Chemicals and Reagents

Commercial powder for injection with one vial delivering 1 g of meropenem and 208 mg of sodium carbonate was from Hikma (London, United Kingdom). Certified reference material of meropenem as the trihydrate, di-potassium hydrogen phosphate trihydrate (K_2_HPO_4_ × 3 H_2_O), and hydrochlorid acid (HCl) were obtained from Supelco (Bellafonte, United States). Commercially available quality controls (QCs) for the therapeutic drug monitoring (TDM) of meropenem in serum were obtained from Chromsystems (Gräfelfing, Germany). Normal saline for intravenous infusion was from B. Braun (Melsungen, Germany) and polypropylene syringes were from H-Medical (Bargteheide, Germany). Drug-free serum was purchased from the blood donation center of the Bavarian Red Cross (Wiesentheid, Germany). The 691 pH meter (Metrohm, Switzerland) was calibrated with standard buffers and used to record pH values of test solutions.

### 4.2. Infusate Stability at Room Temperature

Stability testing in 0.9% normal saline was performed in duplicate. We used only one specific meropenem brand given that generic brands of meropenem were shown to be equivalently stable in normal saline [17]. The antibiotic powder was dissolved in isotonic solutions giving a currently approved dose of 1.0 g meropenem in 50 mL normal saline, yielding a concentration of 2% (corresponding 20 mg/mL). The concentration chosen reflects the standard operating procedure for continuous infusion at the University Hospital, LMU Munich.

In accordance with clinical practice, meropenem continuous infusion solutions were transferred in polypropylene syringes and locked in an infusion pump, delivering with a flow rate of 2 mL/h at 22 °C (±1 °C). At t = 0, 0.5, 1, 2, 4, 6, 8, 10, 12, 14, 16, 20, and 24 h, 200 µL infusates were sampled in polypropylene tubes, immediately frozen in liquid nitrogen, and stored at −80 °C for up to four weeks until ID-LC-MS/MS analysis.

### 4.3. Stability in Serum

Stability testing in serum was performed in duplicate. Drug-free serum was buffered with 3 M phosphate buffer pH 7.4 (K_2_HPO_4_ × 3 H_2_O, titrated with 1 mol/L HCl) (30/1, *v*:*v*) as described by Kratzer et al. [40]. The buffered serum was then warmed to 37 °C (±1 °C) in a water bath and mixed with antibiotic fortified normal saline (95:5, *v*/*v*) prepared from meropenem certified reference material to obtain final concentrations of 2, 10, and 50 mg/L meropenem, respectively. These spiked sera were then aliquoted to 500 µL in polypropylene tubes, incubated at 37 °C (±1 °C) in a water bath. At t = 0, 0.5, 1, 2, 4, 6, 8, 10, 12, 14, 16, 20, and 24 h, aliquots were frozen in liquid nitrogen and stored at −80 °C for up to four weeks until ID-LC-MS/MS analysis. The pH value was measured on replicates and proved to be stable with maximum deviations of +0.3 within these 24 h.

### 4.4. Laboratory Testing and ID-LC-MS/MS Analysis

General clinical chemical parameters were obtained with standard clinical chemical methods. Creatinine clearance was determined using urine collection ((CrCl = (crea_Urin_ × volume_Urin_)/(collection time × crea_Plasma_)) and subsequently used as an estimate for glomerular filtration rate (GFR).

All samples were independently assayed in triplicate with an isotope dilution LC-MS/MS method designed to simultaneously quantify meropenem (molar mass, MM: 383.46 g/mol) and its open-ring metabolite (ORM) (MM: 401.16 g/mol) in a single analytic run [41] (see Figure 6).

In LC-MS/MS analysis, small molecules can be quantified in various biological fluids by measuring the mass-to-charge ratios of target ionized molecular compounds, as well as analyte-specific fragments. Using neat analytes, calibrators and controls are prepared in a given matrix (e.g., serum, plasma) and used for quantitative analysis of matrix-matched samples. To obtain the highest metrological standard, LC-MS/MS is typically combined with isotope dilution standardization. During sample processing, a given concentration of stable isotope-labeled analogues of the analytes of interest is added to a sample (including calibrators and quality controls). These isotope-labeled standards can be selectively detected owing to different molecular weights and fragmentation patterns. Given that they have almost identical physicochemical behavior when compared with their unlabeled counterparts, these standards can be used to efficiently control analytical variations that are introduced from the sample matrix (termed matrix effects). Accordingly, isotope dilution LC-MS/MS has become the gold standard in quantitative small molecule analysis, including TDM [42].

Laboratory-developed calibrators and quality controls (QCs) in both normal saline and serum were obtained from 10x concentrated analyte stock solutions that were prepared by weighing in of meropenem and ORM from certified reference materials.

The linear ID-LC-MS/MS assay range was 1.0–100.0 mg/L for meropenem and 0.62–62.30 mg/L for ORM. Within- and between-run imprecision and inaccuracy were ≤8.0% for all quality controls tested (in most cases, <5%), including four laboratory-developed QCs (with meropenem and ORM) and two commercially available meropenem QC levels. Samples from the continuous infusion stability experiment in normal saline were diluted 1:250 into the linear ID-LC-MS/MS calibration range prior to cleanup. Briefly, 50 µL samples were admixed with internal standard working solution (including meropenem-D_6_ and ORM-D_6_) and precipitated with methanol, and the supernatants were diluted 1:4 in water and separated on a XSelect HSS PFP column (100 × 2.1 mm, 2.5 µm) with a preceding XSelect HSS PFP Van Guard Cartridge from Waters (Milford, MA, USA) with a 10 mM ammonium formiate in water-formic acid (99.8/0.2, *v*/*v*)/methanol gradient elution within 5.5 min. Analysis was done on an Agilent 1290 Infinity I LC system (Santa Clara, CA, USA) coupled to an AB Sciex TripleQuad 6500+ mass spectrometer (Framingham, MA, USA) with multiple reaction monitoring. Specimens from in vitro stability testing were considered stable if solutions retained >90% of the initial concentration at the timepoint t = 0 h [11,12,43].

### 4.5. Calculation of In Vitro Elimination Rate Constants

Graphical and statistical analysis was performed using R Version 4.0.2 (CRAN.R-project.org). A non-linear regression analysis was carried out to determine the first-order elimination rates in perfusor syringes and serum according to the following formula:C(t) = C0 × exp (−t × k)
where k is the elimination rate constant and t is the time.

### 4.6. Patient Serum Testing

Blood samples were collected in the pharmacokinetic steady state once daily in patients with infusion duration >48 h of continuous infusion. Dosing was at the discretion of the responsible physician with a median meropenem infusion rate of 250 mg/h. Patient sera were immediately obtained by centrifugation at 2000× *g* for 10 min at 20 °C and stably stored in polypropylene tubes at −80 °C for up to four weeks until ID-LC-MS/MS analysis. Owing to the negligible protein binding of meropenem, only total meropenem concentrations were considered [44]. Given that the ORM has similar physicochemical properties, it is also not assumed to have significant protein binding.

Only patients aged ≥18 years, admitted to the intensive care unit, treated with continuous infusion of meropenem, and subjected to antibiotic therapeutic drug monitoring (TDM) on at least three consecutive days were included in study. The percentage of ORM was calculated in steady state according to the following formula: Ratio = C_ORM_/(C_ORM_ + C_Meropenem_). Written informed consent was given by all subjects or their legal representatives and the local Ethics Committee of the Ludwig-Maximilians-Universität approved the study (registration number 18–578).

### 4.7. Population Pharmacokinetic Modelling

To compare the calculated elimination rates (see Section 4.3 and Section 4.5) with the total body clearance, a pharmacokinetic model was employed for critically ill patients with continuous infusion. PK modelling was performed using NONMEM^®^ 7.4 (ICON Development Solutions, Ellicott City, MD, USA) and Piraña version 2.9.9 (Certara USA, Inc., Princeton, NJ, USA). A one compartment model with zero-order input and first-order elimination, parameterized in terms of clearance (CL) and volume of distribution (V), was investigated. As all patients were in steady state and the volume of distribution could not be estimated, it was specified according to a published literature value of 26.2 L [26]. The calculated organ independent elimination rate constant in serum was integrated as a fixed value (CL_decay_ = k∗V) into the model, and the renal clearance was fixed to the GFR. Inter-individual variability was incorporated using an exponential model, while residual unexplained variability was investigated using additive, proportional, and combined variability models.

## Figures and Tables

**Figure 1 antibiotics-10-00715-f001:**
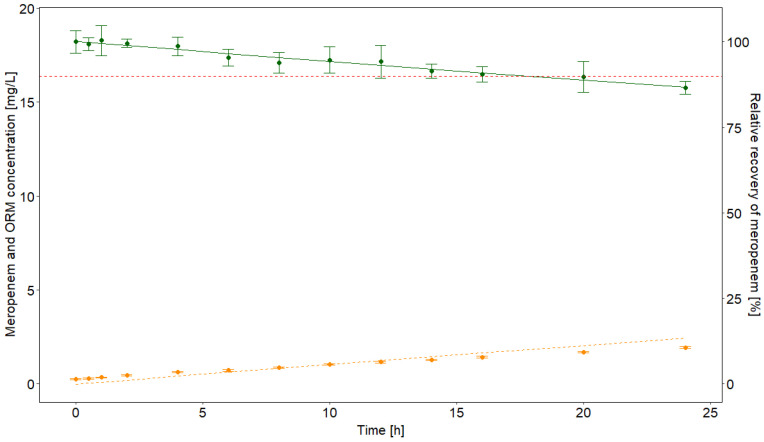
Degradation of 2% meropenem in vitro in normal saline infusate at 22 °C. Red dashed line: 90% relative recovery of meropenem. Green dots: mean measured concentrations of meropenem. Orange dots: mean measured concentrations of the open-ring metabolite (ORM). Standard deviations are depicted with error bars. The non-linear degradation is shown by the green line, while the orange dotted line indicates the theoretical increase in ORM, which is not entirely achieved.

**Figure 2 antibiotics-10-00715-f002:**
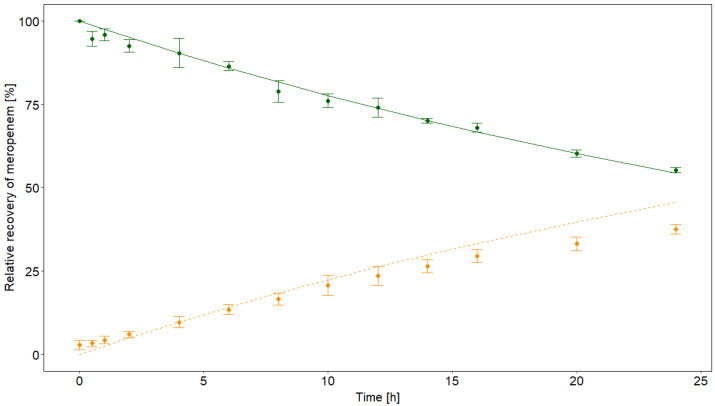
Combined degradation profiles of 2, 10, and 50 mg/L meropenem in vitro in buffered serum at 37 °C. Green dots: mean measured concentrations of meropenem. Orange dots: mean measured concentrations of the open-ring metabolite (ORM). Standard deviations are depicted with error bars. The non-linear degradation is shown by the green line, while the orange dotted line indicates the theoretical increase in ORM.

**Figure 3 antibiotics-10-00715-f003:**
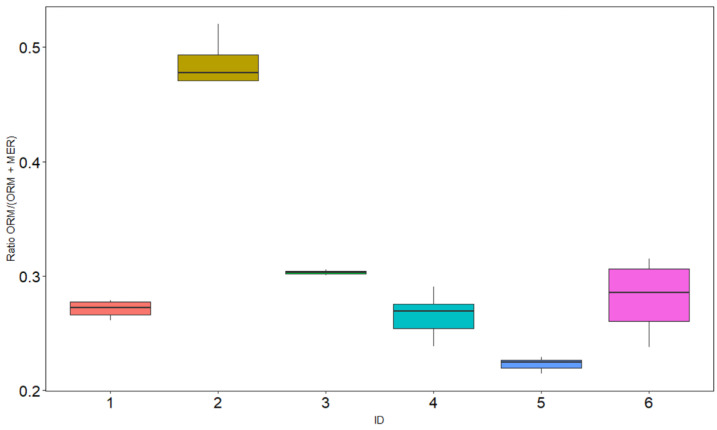
Intra- and interindividual variability of the ratio of the open-ring metabolite (ORM) to the total meropenem concentration in six critically ill patients.

**Figure 4 antibiotics-10-00715-f004:**
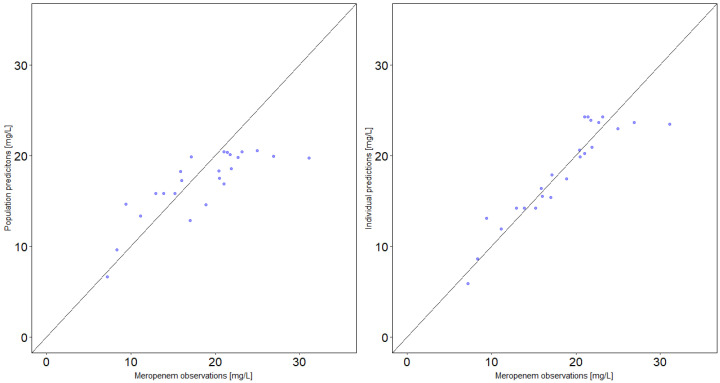
Goodness-of-fit plots for the final population pharmacokinetic model of meropenem in six critically ill patients. Left figure: Population-predicted concentrations versus observed concentrations. Right figure: Individual-predicted concentrations versus observed concentrations. Lines: Line of unity.

**Figure 5 antibiotics-10-00715-f005:**
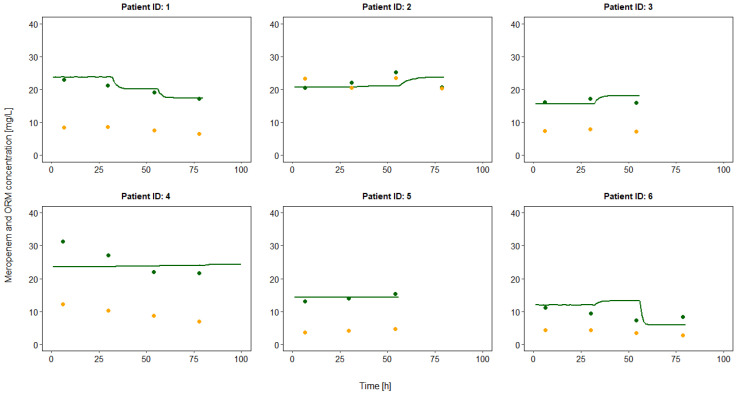
Observed meropenem and open-ring metabolite (ORM) concentrations and meropenem concentration–time profile predicted based on a one-compartment pharmacokinetic model. Green line: median predictions, green points: meropenem concentrations, orange points: ORM concentrations.

**Figure 6 antibiotics-10-00715-f006:**
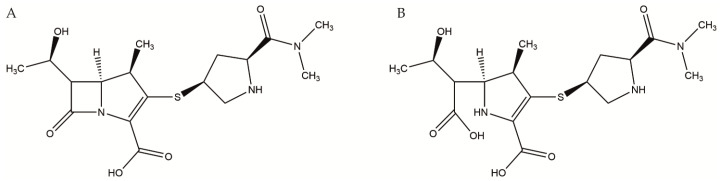
Molecular structure of meropenem (**A**) and its main degradation product, the open-ring metabolite (ORM) (**B**).

**Table 1 antibiotics-10-00715-t001:** Overview of patient characteristics.

Patient Characteristic [Unit]	Number/Median (Range)
No. of patients	6
No. of male patients	4
No. of samples	24
Meropenem concentration [mg/L]	19.74 (7.25–31.25)
Open-ring metabolite (ORM) concentration [mg/L]	7.73 (2.71–23.37)
Meropenem daily dose [g/24 h]	6 (3–6)
Age [years]	46 (35–72)
Weight [kg]	86 (47–170)
Glomerular filtration rate [mL/min]	139 (28–307)

**Table 2 antibiotics-10-00715-t002:** Parameter estimates of the population pharmacokinetic model.

Parameter Estimates (RSE, %) [Shrinkage, %]	
	Meropenem
**Parameter [unit]**	**1-CMT Model**
OFV	121.8
**Fixed-effects Parameter**	
CL_GFR_ [L/h]	7.1
CL_nonGFR_	3.6 (28)
CL_decay_	0.66
V [L]	26.2
**Interindividual variability**	
ω CL (CV %)	14.9 (18) [2]
**Residual variability**	
σ Prop. (CV %)	13.5% (20) [11]

Abbreviations: RSE: relative standard error, CMT: compartment, OFV: objective function value, CL: total clearance, CL_GFR_: clearance attributable to the glomerular filtration rate, CL_nonGFR_: clearance not attributable to the glomerular filtration rate, CL_decay_: clearance attributable to spontaneous decay in plasma at 37 °C (fixed at the experimentally determined value). V: volume of central compartment (fixed at a literature value [26]), ω: random-effects parameters for interindividual variability, CV: coefficient of variation, σ: random-effects parameters for residual variability, Prop.: proportional. Total clearance (CL) was calculated according to the following equation: CL = CL_GFR_ + CL_nonGFR_ + CL_decay_.

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
