# Peer review of "The Role of Non-Enzymatic Degradation of Meropenem—Insights from the Bottle to the Body"

_antibiotics, 2021, doi:10.3390/antibiotics10060715_

Round 1

Reviewer 1 Report

The research carried out in this manuscript is relevant and could be of potential interest for the Antibiotics journal readers. The information is properly presented and the results are adequately shown. In my opinion, the stability test performed for meropenem taking into account the ORM could be valuable in the clinical setting. However, several concerns regarding the methodology applied in the PK analysis must be clarified to better understand specific details of the results presented in this manuscript. The use of PK data of ORM in the analysis is not clear. The meaningful of the PK analysis is not defined in the manuscript (i.e. aims of the manuscript).  In addition, the PK analysis carried out in this manuscript presents a number of limitations that should be acknowledge and discussed.

Please find below specific questions:

INTRODUCTION:

The authors claim the relevance of the inclusion of ORM. In addition, it is highlighted that there are no studies which simultaneously investigate the degradation of meropenem and its conversion to ORM in vivo using a single analytic method. However, the reason of this relevance is missing. I would recommend to include clinical implications of the ORM such as treatment success (efficacy, safety), degradation of the parent drug, etc. to highlight the importance and novelty of this evaluation compared to the previous studies.

Meropenem is mainly excreted by the kidneys. In addition, PKPD targets mentioned in the manuscript are referred to the free drug concentration (unbound concentration). Then, additional information regarding protein binding, drug and RMO unbound concentrations should be discussed.

MATERIALS AND METHODS:

Authors explain a non-linear regression analysis carried out in NONMEM to compare the calculated elimination rates with the total body clearance. However, it seems that two compounds were investigated, meropenem and its metabolite (ORM). Therefore, additional information is required to understand this analysis. For example, how the parent drug and metabolite were evaluated (i.e. simultaneously, sequentially, independently)?

RESULTS:

Lines 114-120: clearance equation must be included in the text or in the footnote of table 1 to allow the reader interpret the values reported.

DISCUSION:

Could the precision and accuracy of the ID-LC-MS/MS analysis (< 8% in all samples) have an impact on the interpretation of the stability tests carried out? Please discuss.

As meropenem PK parameters are estimated and reported in this analysis, the values should be compared with the previous information available in the literature.

Limitations of the study are lacking. The population approach is based on a reduced (6 patients) and specific population (critical ill patients) which has a direct impact on its applicability. In addition, the analysis has several limitations (sparse sampling, fixed parameters, etc.) that must be acknowledge and further discussed.  Authors should be more carefully suggesting to use the PK model developed for dose optimization, specially taking into account the limitations of the analysis performed.

TABLES AND FIGURES

Table 1: please include shrinkage of the random effects.

Figure 4: Please correct caption of figure 4. In addition, only the meropenem concentrations are represented (in the text is indicated as the “pharmacokinetic data”) in this GOF?

Author Response

Comments and Suggestions for Authors

The research carried out in this manuscript is relevant and could be of potential interest for the Antibiotics journal readers. The information is properly presented and the results are adequately shown. In my opinion, the stability test performed for meropenem taking into account the ORM could be valuable in the clinical setting. However, several concerns regarding the methodology applied in the PK analysis must be clarified to better understand specific details of the results presented in this manuscript. The use of PK data of ORM in the analysis is not clear. The meaningful of the PK analysis is not defined in the manuscript (i.e. aims of the manuscript).  In addition, the PK analysis carried out in this manuscript presents a number of limitations that should be acknowledge and discussed.

We thank Reviewer 1 for the thorough revision of our manuscript and suggestions for improvement.

Please find below specific questions:

INTRODUCTION:

The authors claim the relevance of the inclusion of ORM. In addition, it is highlighted that there are no studies which simultaneously investigate the degradation of meropenem and its conversion to ORM in vivo using a single analytic method. However, the reason of this relevance is missing. I would recommend to include clinical implications of the ORM such as treatment success (efficacy, safety), degradation of the parent drug, etc. to highlight the importance and novelty of this evaluation compared to the previous studies.

Meropenem is mainly excreted by the kidneys. In addition, PKPD targets mentioned in the manuscript are referred to the free drug concentration (unbound concentration). Then, additional information regarding protein binding, drug and RMO unbound concentrations should be discussed.

We agree with the reviewer that additional information is needed here and thank the reviewer for this note. Protein binding has been neglected in our analyses as it is negligible low for meropenem. According to the Summary of Product Characteristics the protein binding of meropenem is about 2%, data on critically ill patients confirm this. However, we have taken the liberty of adding this information in the M&M part of the chapter “patient serum testing”. A respective reference and note was included in the manuscript (see line 342). Since ORM has similar physicochemical properties (as it represents hydrolyzed meropenem), it is also not assumed to have significant protein binding. This information was also added to the manuscript (line 341-343).

MATERIALS AND METHODS:

Authors explain a non-linear regression analysis carried out in NONMEM to compare the calculated elimination rates with the total body clearance. However, it seems that two compounds were investigated, meropenem and its metabolite (ORM). Therefore, additional information is required to understand this analysis. For example, how the parent drug and metabolite were evaluated (i.e. simultaneously, sequentially, independently)?

We would like to clarify this point. In a first step, a non-linear regression analysis was performed to calculate the elimination rate constants of the in-vitro experiments. This regression analysis was carried out in R using the following formula: C(t) = C0 • exp (-t • k). Based on this analysis an elimination rate constant of 0.025/h was determined in serum. To investigate the relevance and importance of this elimination process for the total elimination of meropenem (not the ORM), a population pharmacokinetic approach was employed in a second step using NONMEM. The spontaneous decay elimination rate (calculated in Step 1) was inserted in a population pharmacokinetic model, fixed, and subsequently compared to the total clearance.

In order to clarify this procedure for the reader of our manuscript, the section on the calculation of the elimination rate constants (new chapter 4.5, line 327) and the section of the PopPK approach (new chapter 4.7, line 352) were separated in the methods section and altered (also with regards to the improvements raised by reviewer 2 in comment 6). In addition, the benefits of the PopPK approach were substantiated at the end of the introduction (lines 65-67).

 RESULTS:

Lines 114-120: clearance equation must be included in the text or in the footnote of table 1 to allow the reader interpret the values reported.

We added the clearance equation in Table 1 and believe that this note increases the reproducibility of our study (lines 137-138).

DISCUSSION

Could the precision and accuracy of the ID-LC-MS/MS analysis (< 8% in all samples) have an impact on the interpretation of the stability tests carried out? Please discuss.

The guidelines of bioanalytical method validation from the EMA and FDA demand inaccuracy and imprecision values ≤ 15 % for all quality controls, except ≤ 20 % for the lowest QC that has same concentration as the lowest calibrator. Please consider that 8% is the highest deviation in our test series, typically it was ≤ 5 %, which is a good performance for TDM assays. We also emphasize that 6 different quality controls were included per measurement series, although many quantitative assays use only one low and one high control. Accordingly, we are convinced that these measurement uncertainties do not have a significant impact on the interpretation of our data. We now state in the discussion “The inaccuracy and imprecision were always ≤ 8% (in most cases ≤ 5%), which allows reliable conclusions to be drawn.” (line 154-156).

As meropenem PK parameters are estimated and reported in this analysis, the values should be compared with the previous information available in the literature.

We agree with the reviewer that a discussion of the PK parameters is necessary to assess the study content. Discussion of the clearance parameter was included in the discussion section (see lines 201-204). Overall, the total clearance in our model is rather high compared to previously published literature. This can be attributed to a hyperdynamic renal function in our population. To the author’s opinion, the volume of distribution is not relevant for the PK model and the research question of the study (only steady state concentrations were estimated). In addition, it was fixed to a literature value. Accordingly, we believe that it is not useful to discuss this parameter in detail.

Limitations of the study are lacking. The population approach is based on a reduced (6 patients) and specific population (critical ill patients) which has a direct impact on its applicability. In addition, the analysis has several limitations (sparse sampling, fixed parameters, etc.) that must be acknowledge and further discussed.  Authors should be more carefully suggesting to use the PK model developed for dose optimization, specially taking into account the limitations of the analysis performed.

We agree with the reviewer and are happy to take the suggestion. Accordingly, the limitations have been adapted (see lines 219-222).

TABLES AND FIGURES

Table 1: please include shrinkage of the random effects.

Shrinkage was 2 % and added to the corresponding Table (now Table 2). This value is considered adequate by the authors. The statements of the PK model remain unaffected by this.

Figure 4: Please correct caption of figure 4. In addition, only the meropenem concentrations are represented (in the text is indicated as the “pharmacokinetic data”) in this GOF?

Thank you for this very important notification. We sincerely apologize for this lapse of care in drafting the article and thank the reviewer for his attention. The figure caption has been adjusted accordingly. Since the PK model in fact only describes the meropenem concentrations, "pharmacokinetic data" was replaced by "meropenem concentrations" in the text (line 122). We thank the reviewer for the advice and the associated increase in precision of our manuscript.

Reviewer 2 Report

Author's efforts provided more insights into the invivo behavior of meropenem in human volunteers. Although invitro results are well known, the author's research sheds light on the importance of measuring the metabolite exposure levels esp. in a clinical setting with patients having compromised renal function. Manuscript needs a few revisions before it can be deemed publishable. My comments are as below,

1) Please provide the molecular structures of the meropenem and its metabolite ORM. 

2) As the reader base of antibiotics does not largely come from a bioanalytical background, I would suggest authors emphasize more on how isotope dilution analysis was performed and how the concentrations were calculated.

3) Is ORM not available as a synthetic standard? Why did the authors just proceed with meropenem and stable labeled isotopes of meropenem and ORM?

4) The purpose of in vitro infusate experiments is not clear. Stability can be very well measured by leaving the test sample in the syringe, why did the authors perform infusion at 2 mL/min? Also, with an infusion rate of 2 mL/min, how big is the syringe to allow sampling till 24 hours? Overall, I believe the methods part is poorly written. Please enhance the content for better clarity.

5) Please provide an overlaid plasma concentration vs time profile for meropenem and ORM.

6) I would suggest having a separate section with greater details about how the PK study was performed. How the administration was performed, sampling points, and any other details? Are the ratios generated with Cmax or AUC?

7) Please provide a table with demographic info. and other eligibility criteria for all the volunteers?

Overall, the results are interesting but the manuscript has to be organized better esp. the methods part.

Author Response

Comments and Suggestions for Authors

Author's efforts provided more insights into the invivo behavior of meropenem in human volunteers. Although invitro results are well known, the author's research sheds light on the importance of measuring the metabolite exposure levels esp. in a clinical setting with patients having compromised renal function. Manuscript needs a few revisions before it can be deemed publishable. My comments are as below,

We thank Reviewer 2 for the thorough revision of our manuscript and suggestions for improvement.

 1) Please provide the molecular structures of the meropenem and its metabolite ORM.

Molecular structures of both meropenem and its metabolite ORM are now provided in the manuscript as a separate figure (Figure 6) in the M&M section (lines 288-293).

 2) As the reader base of antibiotics does not largely come from a bioanalytical background, I would suggest authors emphasize more on how isotope dilution analysis was performed and how the concentrations were calculated.

Thank you for this valuable comment. We now describe the principle of the isotope dilution standardization in quantitative LC-MS/MS analysis in detail in the material and methods section (line 294-307, 310).

 3) Is ORM not available as a synthetic standard? Why did the authors just proceed with meropenem and stable labeled isotopes of meropenem and ORM?

Indeed, we used both meropenem and ORM reference materials to calibrate our quantitative LC-MS/MS method for both analytes. The application of isotope dilution standardization is considered the highest metrological standard for therapeutic drug monitoring, where usage of isotope labelled analogues of the analytes of interest minimizes sample matrix effects that can interfere with quantitative analysis. Accordingly, we have expanded the M&M part to further explain the principle of isotope dilution standardization in quantitative LC-MS/MS analysis, see previous comment 2.

4) The purpose of in vitro infusate experiments is not clear. Stability can be very well measured by leaving the test sample in the syringe, why did the authors perform infusion at 2 mL/min? Also, with an infusion rate of 2 mL/min, how big is the syringe to allow sampling till 24 hours? Overall, I believe the methods part is poorly written. Please enhance the content for better clarity.

We thank the reviewer for the important comment. We mistakenly stated an infusion rate of 2 mL/min, which was actually 2 mL/h. Accordingly, a 50 mL syringe is sufficient for 24 hours of infusion. We corrected this information in the M&M section, chapter 4.2. Basically, we agree with the Reviewer, that an infusion rate of 2 mL/h would be dispensable and that the reconstituted antibiotic could simply be left in the syringe up to 24 hours and samples manually be drawn at specific time points. Still, we wanted our experimental setup to be as realistic as possible and therefore chose to use an infusion pump that delivers the antibiotic in a typical clinical setting. The method part was fundamentally revised and restructured. In particular, the separation of the paragraph on the calculation of elimination rate constants and the development of the PK-model (see also response to your comment No 6) adds more clarity to our article and increases comprehensibility. In addition, the section 4.6 “patient serum testing” (line 335 – 350) was enhanced.

 5) Please provide an overlaid plasma concentration vs time profile for meropenem and ORM.

Plasma concentration time profiles of meropenem of the six patients were included as Figure 5. The meropenem and ORM measurement points were plotted on top.

6) I would suggest having a separate section with greater details about how the PK study was performed. How the administration was performed, sampling points, and any other details? Are the ratios generated with Cmax or AUC??

We thank the reviewer for pointing this out and have restructured the methods as follows. In detail: Section 4.5. now provides information how the in vitro elimination rates constants were calculated. Section 4.6. is now providing information on the clinical study setting (including information on administration, sampling, inclusion criteria). In addition, we added missing information how the metabolic ratios were exactly calculated: Since all patients were in steady state, the ratio was calculated as follows: Ratio = CORM/(CORM + CMeropenem). Section 4.7 now presents the development of the population pharmacokinetic model.

7) Please provide a table with demographic info. and other eligibility criteria for all the volunteers?

A table providing demographic info was added to the manuscript (now Table 1). Eligibility criteria are now found in section 4.6.

Overall, the results are interesting but the manuscript has to be organized better esp. the methods part.

We have extensively adapted the methods section to make our experiments more comprehensible to a broad readership.

Reviewer 3 Report

Liebchen et al. conducted an in vitro and in vivo pharmacokinetic study of the role of non-enzymatic degradation of meropenem. This is an interesting topic due to the potential implications for patients in the ICU but specially those admitted to OPAT.

Major comments:

The study is interesting, well-written, the objectives are clear, the methods are appropriate and the results are of significance. Although is true that the 12h of stability of meropenem in continuous infusion is already known, authors provide appeling new data on the ORM. Furthermore, the fact that in vivo concentrations have been measured adds a significant value to the study. 

  1. Introduction: OK.
  2. Results: Line 84: Consider to add the point wherer 90 % of degradation is achieved in the text, as has been performed in 2.1.
  3. Discussion: OK
  4. Material and methods: OK

Minor comments:

  1. Line 116: Please modify Pharmakokinetic by pharmacokinetic.
  2. Table 1: Please consider to modify the order of the abbreviations. As a suggestion, authors could follow the order of reading of the table. 
  3. Line 135: This abbreviation has already been spelled before and is not necessary anymore.

Author Response

Comments and Suggestions for Authors

Liebchen et al. conducted an in vitro and in vivo pharmacokinetic study of the role of non-enzymatic degradation of meropenem. This is an interesting topic due to the potential implications for patients in the ICU but specially those admitted to OPAT.

We thank the reviewer for the very positive feedback and the comments that follow.

Major comments:

The study is interesting, well-written, the objectives are clear, the methods are appropriate and the results are of significance. Although is true that the 12h of stability of meropenem in continuous infusion is already known, authors provide appeling new data on the ORM. Furthermore, the fact that in vivo concentrations have been measured adds a significant value to the study.

Introduction: OK.

Results: Line 84: Consider to add the point wherer 90 % of degradation is achieved in the text, as has been performed in 2.1.

Discussion: OK

Material and methods: OK

Thank you for the note concerning the results, chapter 2.2. In the corresponding chapter on serum stability, we have deliberately decided not to include a note on a 90% threshold, given that it refers only to the in vitro stability of infusion solutions according to the EU and US Pharmacopoeia.

 Minor comments:

Line 116: Please modify Pharmakokinetic by pharmacokinetic

Taking into account the comment of another reviewer we changed the term to “meropenem concentrations”.

Table 1: Please consider to modify the order of the abbreviations. As a suggestion, authors could follow the order of reading of the table.

We adapted the order of the abbreviations as suggested.

Line 135: This abbreviation has already been spelled before and is not necessary anymore.

The corresponding abbreviation was removed from the manuscript.

Round 2

Reviewer 1 Report

The authors have adequately response in a point-by-point manner most of the questions / suggestions proposed. They have properly addressed the majority of the concerns raised in the original version of the manuscript improving the work significantly. However, there are still a few questions unclear which might be addressed. The relevance of ORC concentrations on the analysis is still unclear, even it is clear as one of the most significant and innovative point of this research. In addition, number lines of the answers do not match with the ones in the reviewed manuscript.

Please find below specific questions:

INTRODUCTION:

Could the authors explain the relevance of ORC information to the analysis (e.g. efficacy, safety, prediction factor of meropenem degradation, etc.) to better understand the importance and novelty of this evaluation compared to the previous studies?

MATERIALS AND METHODS:

Section 4.5. Authors refers to a non-linear regression analysis from an in vitro experiment. Afterwards, they claim to input this elimination rate constant (ke) value to the NONMEM analysis:

  • I would recommend to provide additional information on how the in vitro experiment was performed to obtain this data.
  • In addition, is there any benefits to input the Ke obtained from an in vitro experiment rather to be estimated from the “NONMEM analysis”?
  • Finally, was the ORC metabolite included in the meropenem elimination pathways? Did the authors take into account the ORC concentrations as one of the meropenem possible elimination pathways?

DISCUSION:

Lines 219-221: “Total clearance in our population was slightly increased compared to previously published meropenem models in critically ill patients, which can be attributed to an overall hyperdynamic renal function (median GFR 139 mL/min)”; Please include CL range of previous reported values in the literature.

TABLES AND FIGURES

Table 1: the authors included the equation of CL as follows: CL= CLGFR + CLnonGFR + CLdecay; Could the authors that the term CLdecay is an additive component contributing to the amount of the total CL? Due to the name assigned (“decay”) and looking at the data, it would be more reasonable to be a proportional component which decays the value of CL. Could the authors cross check this point?

Table 2: please include shrinkage of residual unknown variability.

Author Response

Comments and Suggestions for Authors

The authors have adequately response in a point-by-point manner most of the questions / suggestions proposed. They have properly addressed the majority of the concerns raised in the original version of the manuscript improving the work significantly. However, there are still a few questions unclear which might be addressed. The relevance of ORC concentrations on the analysis is still unclear, even it is clear as one of the most significant and innovative point of this research. In addition, number lines of the answers do not match with the ones in the reviewed manuscript.

We thank again Reviewer 1 for the thorough reading and helpful remarks. I am sorry for the incorrectly displayed lines. The docx-manuscript file was uploaded with the changes introduced with the “track changes” option. However, upon submission all these changes were permanently incorporated by the editorial team and only marked in yellow, which is why the line numbers had changed.

Please find below specific questions:

INTRODUCTION: 

Could the authors explain the relevance of ORC information to the analysis (e.g. efficacy, safety, prediction factor of meropenem degradation, etc.) to better understand the importance and novelty of this evaluation compared to the previous studies?

As suggested, we now specify a further added value for the ORM in the introduction (lines 59-62). Given that meropenem itself is rather unstable, a high ORM concentration could be indicative of a pre-analytical error (e.g. in vitro meropenem degradation).

MATERIALS AND METHODS:

Section 4.5. Authors refers to a non-linear regression analysis from an in vitro experiment. Afterwards, they claim to input this elimination rate constant (ke) value to the NONMEM analysis:

I would recommend to provide additional information on how the in vitro experiment was performed to obtain this data.

The detailed description of how the experiments mentioned were conducted can be found in our manuscript in sections 4.3 (experimental design), 4.5 (statistical evaluation) and section 2.2 (results). To make this related information more visible, section 4.3 has been referenced additionally in section 4.7 (line 336). We hope that this will allow the readership to easily trace the experimental determination of the elimination rate in human serum.

In addition, is there any benefits to input the Ke obtained from an in vitro experiment rather to be estimated from the “NONMEM analysis”?

We would like to address this fundamental question. In principle, the non-renal elimination could also be estimated, as it has already been done in preliminary work (e.g. DOI: 10.1186/s13054-018-1940-1). However, such estimates contain a considerable uncertainty and are often not backed up by sufficient data quality. In such estimates, a distinction can be made between glomerular filtration and other elimination pathways, but a further differentiation for the non-glomerular filtration pathway is not possible. The incorporation of the experimentally determined decay-rate thus provides a well-grounded data basis for the differentiation of spontaneous decay and other elimination pathways that cannot be attributed to glomerular filtration. We therefore believe that the integration of an experimentally based constant is a clear benefit to a mere estimation.

Finally, was the ORC metabolite included in the meropenem elimination pathways? Did the authors take into account the ORC concentrations as one of the meropenem possible elimination pathways?

We would like to respond to the reviewer's suggestion to determine non-renal clearance via the amount of ORM. The total clearance is to be regarded as a constant, in our model it is 11.4 L/h (and can be easily calculated without NONMEM or other advanced pharmacometric software: Css=rate of infusion/clearance). In order to determine the relevance of the sub-elimination pathways (glomerular filtration, spontaneous decay, and other non-renal elimination pathways), quantitative sub-values of the clearance must necessarily be fixed or calculated based on data. Two possibilities now arise: 1) as the reviewer suggests, one could estimate the total non-renal clearance via the amount of ORM (e.g. by including a second compartment). However, using this approach it is not possible to distinguish between spontaneous decay and cleavage via the DHP, as the same product is formed. The remaining part of the total clearance would then be accounted for by glomerular filtration. 2) Since glomerular filtration is known in our study (urine creatinine clearance), glomerular filtration can be fixed and the remaining residual clearance can be estimated and represents the clearance not attributable to glomerular filtration. The information gained thus remains the same in both scenarios.

In both cases, a distinction between glomerular filtration and other pathways can be made. The relationship to spontaneous decay can only be reliably estimated by integrating the experimentally determined decay rate. Overall, we believe that the reviewer's suggestion is as good an option as our chosen approach, but no additional information can be derived from it.

DISCUSION:

Lines 219-221: “Total clearance in our population was slightly increased compared to previously published meropenem models in critically ill patients, which can be attributed to an overall hyperdynamic renal function (median GFR 139 mL/min)”; Please include CL range of previous reported values in the literature.

We have gladly included the missing information in our manuscript (line 200) and are convinced that this information will make our work more precise.

TABLES AND FIGURES

Table 1: the authors included the equation of CL as follows: CL= CLGFR + CLnonGFR + CLdecay; Could the authors that the term CLdecay is an additive component contributing to the amount of the total CL? Due to the name assigned (“decay”) and looking at the data, it would be more reasonable to be a proportional component which decays the value of CL. Could the authors cross check this point?

We thank the reviewer for this critical analysis. However, we would like to point out why we have chosen to use an additive term: to the best of our knowledge, the elimination of meropenem predominantly occurs in the kidneys (glomerular filtration, to a lesser extent also secretion and enzymatic cleavage by the DHP). The non-enzymatic decomposition/decay, which was experimentally tracked in our study, takes place in parallel in the entire body and independently of renal processes. A proportional consideration of spontaneous decay would indicate that the non-enzymatic reaction would be dependent on renal excretion, which, according to our understanding, would not correspond to physiological processes. Since the proportion of non-renal elimination actually increases with decreasing renal function, a proportional consideration is therefore misleading.

Table 2: please include shrinkage of residual unknown variability.

Shrinkage was included according to the reviewer’s suggestion in Table 2 and was 11 %.
